# NEURAL ARITHMETIC UNIT BY REUSING MANY SMALL PRE-TRAINED NETWORKS

## ABSTRACT

We propose a solution for evaluation of mathematical expression. However, instead of designing a single end-to-end model we propose a Lego bricks style architecture. In this architecture instead of training a complex end-to-end neural network, many small networks can be trained independently each accomplishing one specific operation and acting a single lego brick. More difficult or complex task can then be solved using a combination of these smaller network. In this work we first identify 8 fundamental operations that are commonly used to solve arithmetic operations (such as 1 digit multiplication, addition, subtraction, sign calculator etc). These fundamental operations are then learned using simple feed forward neural networks. We then shows that different operations can be designed simply by reusing these smaller networks. As an example we reuse these smaller networks to develop larger and a more complex network to solve n-digit multiplication and n-digit division. This bottom-up strategy not only introduces reusability, we also show that it allows to generalize for computations involving n-digits and we show results for up to 7 digit numbers. Unlike existing methods, our solution also generalizes for both positive as well as negative numbers.

## 1 INTRODUCTION

The success of feed-forward Artificial Neural Network (ANN) lies in their ability to learn that allow an arbitrarily connected network to develop an internal structure appropriate for a particular task. This learning is dependent on the data provided to the network during the training process. It has been commonly observed that almost all ANNs lack generalization and their performance drastically degrades on unseen data. This includes degradation of performance on data containing the seen categories but acquired under from a different setup (location, lighting, view point, size, ranges etc). Although there are techniques such as Domain Adaptation to address these generalization issues, however this behaviour indicates that the learning process in neural network is primarily based on memorization and they lack understanding of inherent rules. Thus the decision making process in ANN is lacking quantitative reasoning, numerical extrapolation or systematic abstraction.

However when we observe other living species, numerical extrapolation and quantitative reasoning is their fundamental capability what makes them intelligent beings. For e.g. if we observe the learning process among children, they can memorize single digit arithmetic operation and then extrapolate it to higher digits. More specifically our ability to $+$, $-$, $\times$ and $\div$ higher digit number is based on understanding how to reuse the examples that we have memorized for single digits. This indicates that the key to generalization is in understanding to reuse what has been memorized. Furthermore, complex operations are usually combination of several simple function. Thus complex numerical extrapolation and quantitative reasoning among ANNs can be developed by identifying and learning the fundamental operations that can be reused to develop complex functions.

Inspired from the methodology of learning adopted by humans, in this work we first identify several fundamental operations that are commonly used to solve arithmetic operations (such as 1 digit multiplication, addition, subtraction, merging of two number based on their place value, learning to merge sign $+/-$ etc). These fundamental operations are then learned using simple feed forward neural networks. We then reuse these smaller networks to develop larger and a more complex network to solve various problems like n-digit multiplication, n-digit division, cross product etc. To the best of our knowledge this is the first work that proposed a generalized solution for these arith-

metic operations. Furthermore, unlike exiting methods ( Hornik et al. (1989); Siu & Roychowdhury (1992); Peddada (2015); Sharma (2013); Trask et al. (2018)) ours is also the first solution that works for both positive as well as negative numbers.

## 2 RELATED WORK

Neural Network are known for their ability to approximate mathematical functions (see Hornik et al. (1989)). Exploratory work has been done to approximate simple arithmetic operations which include multiplication and division (see Siu & Roychowdhury (1992)). Although, it might seem a fairly intuitive task but the inability to generalize over unseen data makes the architecture of proposed networks complex in nature (see Cho et al. (2019)). Recent works including Resnet (He et al., 2016) through identity mappings, highway networks (Srivastava et al., 2015) through gating units, and dense networks Huang et al. (2017) through densely connected layers; have tried to train networks which can generalize over minimal training data. To this effect, EqnMaster (see Peddada (2015)) uses generative recurrent networks to approximate arithmetic functions (addition, subtraction and multiplication only). However, this model still does not generalize well as it is only able to perform these functions over 3-digit numbers. Most recent work in this regard is Neural Arithmetic Logic Unit (NALU) (see Trask et al. (2018)) which uses linear activations to represent numerical quantities and then using the learned gate operations predicts the output of arithmetic functions. The results of NALU (Trask et al. (2018)) determine that extrapolation issues will always be a problem in end-to-end learning based tasks. A simple Feed Forward Network (see Franco & Cannas (1998)) can determine solutions to arithmetic expressions including multiplication. This approach does not result in the most efficient network architecture to solve even the simplest arithmetic expression (for example: 3-depth network for multiplication). Optimal Depth Networks (see Siu & Roychowdhury (1992)) through a deterministic approach using binary logic gates can perform simple arithmetic functions. Using digital circuits as an inspiration for neural network architecture has been studied in various researches including Digital Sequential Circuits (se Ninomiya & Asai (1994)) and Simulation of Digital Circuits (see Sharma (2013)). Using digital circuitry as a point of reference neural networks can be designed to solve simple arithmetic problems. Our work establishes on the premise provided by Binary Multiplier Neural Networks (see Huang et al. (2013)) that perform arithmetic operations on binary digits. We propose a network that can deterministically predict the output of basic arithmetic functions over the set of both $n$-digit positive and negative decimal integers as opposed to the state-of-the-art model which only works on (limited digits) positive integers.

These existing methods that propose a single neural network that when trained independently for different tasks attempt to predict output for different functions. We instead propose to train several smaller networks to perform different sub tasks that are needed to complete a difficult arithmetic operation such as signed multiplication, division and cross product. We then use various combination of these already trained smaller networks and design networks that can perform complex task. Similar, to LSTMs, we also propose a loop unrolling strategy that allows to generalize the solution from seen 1 digit arithmetic to unseen $n$-digit arithmetic operations.

## 3 FUNDAMENTAL OPERATION

Multiplication is often performed as repeated addition similarly, division can be performed as repeated subtraction( ()). These operations are implemented on digital circuits as shift and accumulator to sum each partial products. These digital circuits are know to perform accurate arithmetic operations and are commonly used in digital computing equipment. They can also be easily scaled by increasing the shifters and accumulators( ()). Furthermore, some initial work has been done to shown that neural networks can be used to simulate digital circuits (Ninomiya & Asai (1994)).

Using this as inspiration we analyze that $n$-digit multiplication on paper and pencil involve 1-digit multiplication, addition, place value shifter, and sign calculator. Division operation intern requires subtraction operation in addition to those needed for multiplication. We thus designed 6 neural networks to perform 8 fundamental operations namely i) addition, ii) subtraction iii) 1-digit multiplication iv) place value shifter, v) input sign calculator, vi) output sign calculator, vii) 1 digit sign multiplication and viii) absolute. We then design several complex neural network one for each com-

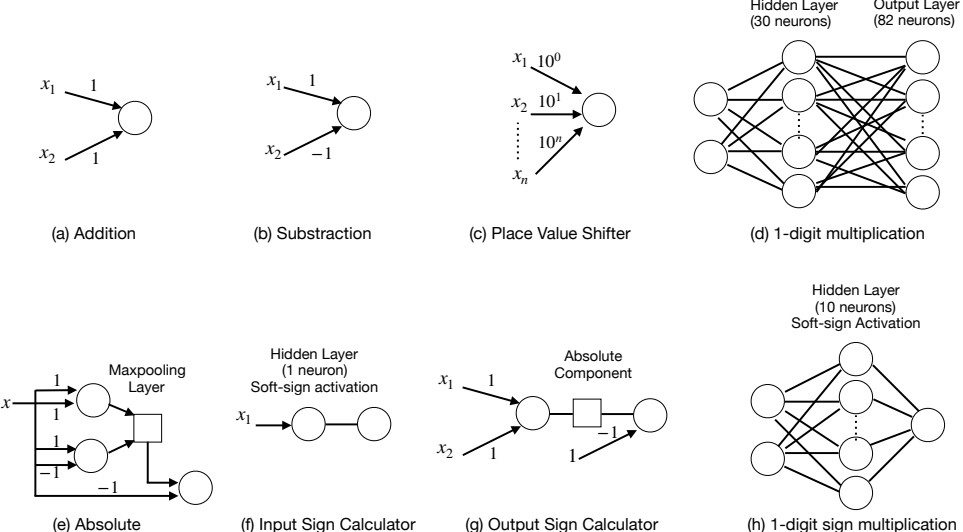

Figure 1: Fundamental operations

plex function such as multiplication and division and show that these fundamental blocks can be used to design several complex functions including a arithmetic equation calculator.

### 3.1 ADDITION AND SUBTRACTION

The basic function of a neuron network is a sum transform where each input is multiplied with a weight, these weighted inputs are then passed through an activation function to produce the final output. This inherent summation property simplifies the task of creating a neural network addition of two numbers. Our addition module is thus implemented using a single neuron (having linear activation) with two inputs by setting their weights to $\{+1, +1\}$. Similarly, our subtraction module consist of a single neuron with two inputs by setting their weights to $\{+1, -1\}$ (see Fig. 1(a-b)).

### 3.2 PLACE VALUE SHIFTER

This module is designed to facilitate shift-and-add multiplication in which each digit of the multiplier is multiplied one at a time from right to left with each digit of the multiplicand. The output of these multiplications are then combined to create a single number. The place value shifter is used to place the output at the appropriate position followed by addition to obtain the final output. Similar to addition network, this module is also composed of a single neuron with linear activation. Instead of taking only two integer values, it can be unrolled for $n$ inputs each taking a 1 digit number as input. This network uses fixed weights in the power of 10 for each preceding digit (see Fig. 1(c)).

### 3.3 1-DIGIT MULTIPLICATION

In 1-digit integer multiplication there are 82 possible outcomes (digits 1 through 9 multiplied with each other and outcome of multiplication with 0). Thus our proposed feed forward network to learn the results of single digit multiplication has two input neurons, 1 hidden layer with 30 neurons and an output layer of 82 neurons. The model takes as input two 1-digit integers and produces 82. The highest-ranked prediction is then selected as the output of single digit multiplication (see Fig. 1(d)).

### 3.4 ABSOLUTE

This network takes a single number and computes its absolute value using a neural network having 2 hidden layer. The first hidden layer performs the $x + x$ and $x - x$ operation, the second layer is a

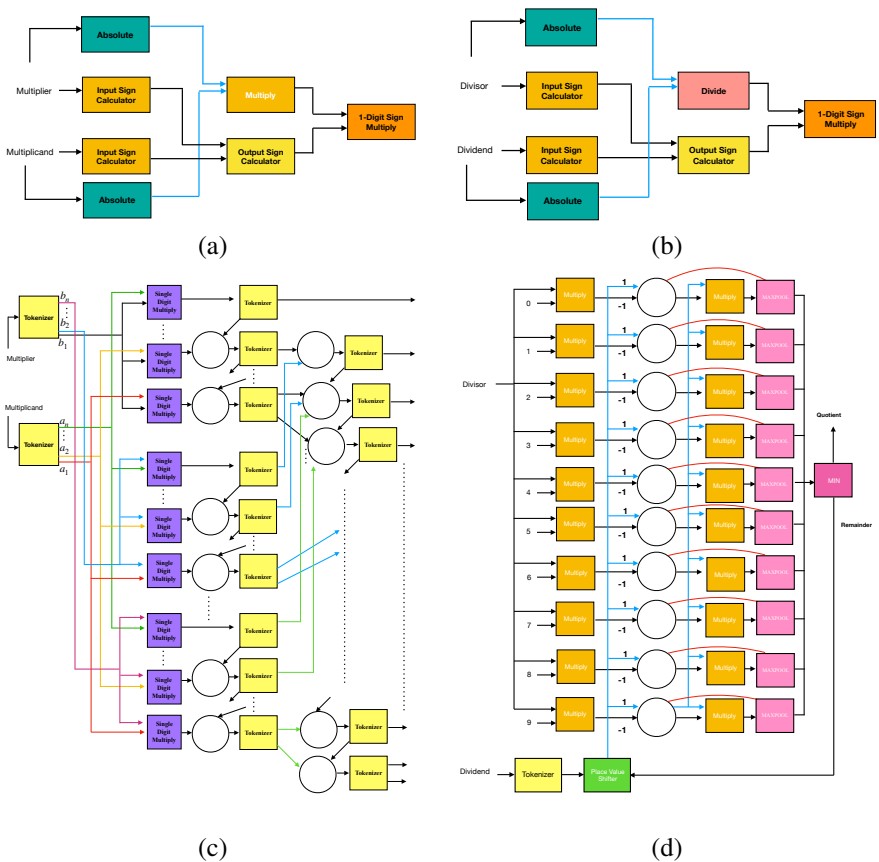

Figure 2: Multiplication and Division using fundamental operations. (a) and (b) are the high-level representation of signed multiplication and signed division operations respectively. (c) and (d) represents the unrolled architecture for $n$-digit multiplication and $n$-digit division (respectively) of two positive integers.

maxpool layer and the final output layers subtracts the input from the output of maxpool layer. The entire process can be represented as $max(x + x, x - x) - x$ (see Fig. 1(e)).

## 3.5 INPUT/OUTPUT SIGN CALCULATOR

The input sign calculator is used to extract sign of an input number $x$. It is a single neuron that uses the following activation function $x/(1 + mod(x))$ (see Fig. 1(f)).

The output sign calculator computes the resultant from a multiplication or division of two numbers. It takes two inputs which are a combination of 1 and $-1$ $(1, 1, 1, -1, -1, 1, -1, -1)$ and generates the resulting sign. The operation is modelled as neural network with 2 hidden layers. The first layer simply adds the two numbers, the second layer is a single neuron that uses $modulus$ as its activation function and the final layer performs a subtraction of 1 from the output of previous layer. The process can be represented mathematically as $mod(x1 + x2) - 1$ (see Fig. 1(g)).

## 3.6 1-DIGIT SIGN MULTIPLICATION

The network takes as input sign and magnitude which is passed on to a hidden layer of 10 neurons. The output layer has a soft-sign activation to predict the output of the sign multiplication. The module is used to assign a sign to the output of a complex operation such as multiplication and division (see Fig. 1(h)).

## 4 MULTIPLICATION

We perform multiplication between $2$ $n$ digit signed integers by using all the $8$ different fundamental operations. The high-level diagram of signed multiplication is shown in Fig. 2(a). First the two numbers are converted into positive integers by using the absolute operation. The input and output sign calculators are also used here to extract the sign of input numbers and to compute sign of resulting output value. The multiplication between the two positive integers obtained via absolute operation is then performed using a multiply sub module shown in Fig. 2(c).

The multiply sub module tokenizes the inputs into single digits. The single digit multiplication operation is then used to multiply each token of multiplicant with the 1-st token of multiplier. The output of each of the these operations is then added with the carry forward from the 1-digit multiplication with the previous token. Each of these results are then combined to form a single number. This process then repeats for the each token of the multiplier. The final output is then assigned a sign using 1-digit sign multiply (see also Algorithm 1.

---

**Algorithm 1** Multiplication model based on digital circuitry

---

m-digit multiplicand and n-digit multiplier
Expected number of output digits for an n*m multiplication are n+m digits.
Total elements produced by row-wise multiplication = (m+1)*n
Each row can have a maximum of m+1 output digits.

$list = []$
$final\_Output = []$
$current\_node = first\ row\ first\ output$
**for** *output node:* **do**
    create $new\_node = current_node$

    $next\_node = mth\ node\ from\ current\ node$

    **loop**
        create $new\_node = next\_node$ parallel to $current\_node$

        $next\_node = mthnode\ from\ current\_node$

    **if** $next\_noderesults\ in\ out\text{-}of\text{-}bound\ index$ **then**
        break
    **end**

    **if** $next\_node$ *meets the base case* **then**
        break
    **end**

    **end loop**
    SUM = use Adder network to find the sum of all nodes in the list

    (Ones, Tens) use Separator (look Figure (D)) to separate ones and tens

    list = [Tens]

    append Ones to the $final\_Output$

    $current\_node$ is set to the next output node
**end**

---

## 5 DIVISION

The division model also separates the sign and magnitude during pre-processing. The architecture is inspired from long division model where n-digit divisor controls the output computation. The n-digit divisor is multiplied with single digit multipliers (0 through 9) and subtracted from selected n-digit chunk of dividend. From the given set of outputs we select smallest non-negative integer by introducing two additional layers for this specific purpose. The selected node represents the remainder and the quotient result of division for the selected chunk of n-digit dividend and n-digit divisor. The quotient is combined over all iteration and the remainder is knitted to the next digit available in the divisor. Figure 2(b,d) shows the architecture of multiplication network.

We can generate a division model based on digital circuitry for decimal digits as shown in Algorithm 2.

---

**Algorithm 2** Division model based on digital circuitry

---

$Inputs = (divisor, dividend)$
$Quotient = []$
$Remainder = []$
$length\_of\_divisor = n$
$dividend\_index_s tart = 1$
$dividend\_index_e nd = n$
$X = Divisor. * 0, 1, 2, 3, 4, 5, 6, 7, 8, 9$
$current\_dividend = dividend[dividend\_index\_start : dividend\_index\_end + 1]$
**for do**

   $Y1 = current\_dividend - X$

   $Y2 = Y1. * Y1[10]$

   $Y = []$

   **for** *i:= 1 to 10* **do**
   $\quad$ append max(Y1i, Y2i) to Y
   **end**

   $(output, index) = (min(Y), min_a t(Y))$

   append index to Quotient

   $dividend\_index_s tart = dividend\_index\_end + 1$

   $dividend\_index_e nd = dividend\_index\_end + 2$

   $current\_dividend = [output, dividend[dividend\_index\_start : dividend\_index\_end]]$

   **if** $dividend\_index\_end$ *is out of bounds* **then**
   $\quad$ break
   **end**

**end**

$Remainder = current\_dividend$

Merge Quotient

---

Table 1: Comparison of proposed model with existing deep networks showing classification accuracy in percentage.

|         | D-RNN | D-GRU | BB   | RNN  | GRU  | LSTM | R-RNN | Ours |
|---------|-------|-------|------|------|------|------|-------|------|
| $+$     | 89.2  | 95.8  | 71   | 86.5 | 97.7 | 97   | 95.1  | 100  |
| $-$     | 93.4  | 95.6  | 51.4 | 82.8 | 96.6 | 93   | 94.6  | 100  |
| $\times$| 92.6  | 81.3  | 36.5 | 13.5 | 51.1 | 40.9 | 41.3  | 100  |

Table 2: Comparison with Neural Arithmetic and Logic Unit (NALU) (Trask et al., 2018). This comparison is only values between 0-30 as this is the range in which NALU is trained and operates.

|      | $+$ | | $-$ | | $\times$ | | $\div$ | |
|------|-----|-----|-----|-----|-----|-----|-----|-----|
|      | Int | Exp | Int | Exp | int | Exp | Int | Exp |
| NALU | 100 | 100 | 93.1 | 95 | 100 | 97.2 | 98.9 | 97.2 |
| Ours | 100 | 100 | 100 | 100 | 100 | 100 | 100 | 100 |

## 6 EXPERIMENTS

### 6.1 SETUP

We compare our model with the results provided by Peddada (2015) on their dataset. Since, they only implement addition, subtraction and multiplication so, we can not compare the results for division architecture proposed in this paper. We do however, compare the result of signed arithmetic operations.

For our second comparison, we have used the implementation provided by Neural Arithmetic and Logic Unit (NALU) Trask et al. (2018) and train it to match the results claimed by them. Once trained, we use their prediction dataset (range: 0 to 30 uniform numbers) and round them to integer values. We calculate the test results on this data for NALU and our implementation.

To support our claim of generalization we test our model on 2-digit integers up to 7-digit integers which are randomly generated. The dataset also includes negative integers to check the accuracy for signed arithmetic operations.

### 6.2 RESULTS

**Experiment 1**: This experiment shows that our model outperforms the recurrent and discriminative networks (Peddada, 2015). Even within the testing range of their dataset we get $100\%$ accuracy. Furthermore, the signed multiplication is exclusive to our model only. Table 1 shows the results of this comparison.

**Experiment 2**: We compares our results to the state-of-the-art model for arithmetic operations on positive integer values called Neural Arithmetic and Logic Unit (NALU) (Trask et al., 2018). This experiment also allows us to compare our results for the division architecture proposed in our paper. The NALU network is only trained between the range of $10 - 20$ outside which it fails drastically. For fair comparison we have used their generated test set in this experiment. Table 2 shows the results of this comparison.

**Experiment 3**: We also conducted experiments to demonstrate that our model is able to generalize to $n$-digits for signed arithmetic operations (see Table 3). It scales in size with the complexity of the inputs and performs exactly the same on all inputs. After the learning for the atomic components our model demonstrates output accuracy is data independent.

Table 3: Results showing generalization of our proposed approach on higher digit integers

|  | Stanford Dataset | | | Inhouse Dataset | | | |
|---|---|---|---|---|---|---|---|
|  | + | − | × | + | − | × | ÷ |
| 2 digit | 100 | 100 | 100 | 100 | 100 | 100 | 100 |
| 3 digit | 100 | 100 | 100 | 100 | 100 | 100 | 100 |
| 4 digit | 100 | 100 | - | 100 | 100 | 100 | 100 |
| 5 digit | 100 | 100 | - | 100 | 100 | 100 | 100 |
| 6 digit | 100 | 100 | - | 100 | 100 | 100 | 100 |
| 7 digit | 100 | 100 | - | 100 | 100 | 100 | 100 |

## 7 CONCLUSION

In this paper we show that many complex tasks can be divided into smaller sub-tasks, furthermore many complex task share similar sub-tasks. Thus instead of training a complex end-to-end neural network, many small networks can be trained independently each accomplishing one specific operation. More difficult or complex task can then be solved using a combination of these smaller network. In this work we first identify several fundamental operations that are commonly used to solve arithmetic operations (such as 1 digit multiplication, addition, subtraction, place value shifter etc). These fundamental operations are then learned using simple feed forward neural networks. We then reuse these smaller networks to develop larger and a more complex network to solve various problems like n-digit multiplication and n-digit division.

One of the limitation of the proposed work is the use of float operation in the tokenizer which limits the end-to-end training of complex networks. However, since we are only using pre-trained smaller network representing fundamental operations, this does not creates any hinderance in our current work. However, we aim to resolve this issue in future. We have also designed a cross product network using the same strategy and we are currently testing its accuracy. As a future work we aim to develop a point cloud segmentation algorithm by using a larger number of identical smaller network (i.e. cross product) that can compute a normal vector using 3 3D points as input.

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
