# OpenReview forum: "Neural Arithmetic Unit by reusing many small pre-trained networks"
_ICLR.cc/2020/Conference — Reject_

### Official Review · AnonReviewer2 · 2019-10-21
**Official Blind Review #2**

**Rating:** 1

**Review:**

This paper proposes to use neural networks to evaluate the mathematical expressions by designing 8 small building blocks for 8 fundamental operations, e.g., addition, subtraction, etc. They then design multi-digit multiplication and division using these small blocks.

The motivation of this paper is not very clear to me, i.e., why do you want to mimic the arithmetic operations using the logic networks, what is the real use case here. In the introduction, the paper motivates by pointing out the limitation of neural networks, which is memorization based and they want to generalize by understanding the inherent rules. However, if you look at the way the fundamental building blocks are designed, and how the multiplication model works, the rules are injected based on human knowledge, e.g., the way signal digit multiplication extends to multi-digit multiplication, there is simply no understanding by the model itself. Besides, the whole process has no training, i.e., the weights of the small networks are fixed, and what is the trainable parts?

The whole paper has many spelling and grammar errors, which hinders the reading. And the writing needs to be significantly improved.

**Experience Assessment:**

I do not know much about this area.

**Review Assessment: Checking Correctness Of Derivations And Theory:**

I assessed the sensibility of the derivations and theory.

**Review Assessment: Checking Correctness Of Experiments:**

I assessed the sensibility of the experiments.

**Review Assessment: Thoroughness In Paper Reading:**

I read the paper thoroughly.

---

### Official Review · AnonReviewer1 · 2019-10-22
**Official Blind Review #1**

**Rating:** 1

**Review:**

The paper proposes a method to design a NN based mathematical expression evaluation engine. I find that the paper could benefit a lot from some rewriting as it is not very clear and over claiming at points.
The introduction states that almost all ANNs lack generalization, this is in my opinion an overstatement. Domain shift and adaptation are techniques to cope with situations where the test data distribution is not coherent with the training data distribution. If this would be true in general we would have not seen such a resurgence and widespread use of ANN in the past years.

The paper lacks also proper citations to previous work and I find the background section and motivation rather weak.

The fundamental operations presented in section 3 do not involve any learning at all, contrary to the referenced work of Trask et al where parameters are actually learned (see relaxation of sign function with tanh etc.). I therefore find the use of ANN as basic constituent of the block to be wrong, each network has fixed hand-crafted weights. If I were to replace ANN with the ordinary corresponding function nothing would in the presented framework.

Multiplication and division as explained in the algorithms do not require learning at all. I am afraid the ML contribution of this work is in my opinion almost non existent. I see every component as being scripted rather than learned from the data, which would be of course much more interesting.

Experiments are not clear at all, setup and explanation of results are not sufficient and I find them not thoroughly executed. Table 1 mentioned classification but the task is never clearly explained. Experiment 2 compares to NALU but in the proposed work nothing is learned, unless I misunderstood the work entirely.



**Experience Assessment:**

I have read many papers in this area.

**Review Assessment: Checking Correctness Of Derivations And Theory:**

I assessed the sensibility of the derivations and theory.

**Review Assessment: Checking Correctness Of Experiments:**

I carefully checked the experiments.

**Review Assessment: Thoroughness In Paper Reading:**

I read the paper at least twice and used my best judgement in assessing the paper.

---

### Official Review · AnonReviewer3 · 2019-10-26
**Official Blind Review #3**

**Rating:** 1

**Review:**

Writing
Overall, the readability of this paper is far from the acceptance criteria of ICLR. there are just way too many grammatical errors or typos throughout the entire paper that prevent me from understanding this paper, such as
Sec.1
Although…, however…
In neural network -> neural networks
They lack understanding -> it lacks understanding
Numerical and quantitative reasoning is their r fundamental capability -> are .. capabilities…
...
Just too many to print all of them here. Please proofread your paper before submission.

The introduction is poorly written that I cannot get a full picture of what goals this paper tries/has achieved after reading it.

Algorithm 1 and 2 seem to be very poorly formatted but only illustrate minimal useful information.


Motivation
I don’t see clear usage nor convincing results based on the current shape of this paper -- what application could this work enable or what theoretical insights it reveals?

Method
This paper proposes to use neural nets to do arithmetic operations (though I don’t see convincing motivations to do so). A new idea the paper proposes is to train a few networks to first learn/fit basic operations, and then use these trained NNs to assemble large NNs which are supposed to form more complex arithmetic operations. Unfortunately, the writing of this paper prevents me from fully understanding the technical details of this paper.

Results
The results in this paper are currently minimal. Many details about the experiment setup or how different methods are compared are not clear.


**Experience Assessment:**

I have read many papers in this area.

**Review Assessment: Checking Correctness Of Derivations And Theory:**

I did not assess the derivations or theory.

**Review Assessment: Checking Correctness Of Experiments:**

I assessed the sensibility of the experiments.

**Review Assessment: Thoroughness In Paper Reading:**

I read the paper at least twice and used my best judgement in assessing the paper.

---

### Decision · Program_Chairs · 2019-12-19

**Decision:**

Reject

**Comment:**

This paper proposes to train and compose neural networks for the purposes of arithmetic operations. All reviewers agree that the motivation for such a work is unclear, and the general presentation in the paper can be significantly improved. As such, I cannot recommend this paper in its current state for publication.